# Prevalence of *Campylobacter* and non-typhoidal *Salmonella* along broiler chicken production and distribution networks, Northern Vietnam

Chen Xin[1,2]*, Thuy Thi Hoang[3], Duong Quy Truong[3], Nhat Thi Tran[3], Hoa Thi Thanh Pham[4], Son Thi Thanh Dang[3], Huong Quynh Luu[3], Nhung Thi Hong Le[3], Saira Butt[5], Hang Thi Thu Tran[3], Kelyn Seow[6], Mathew Hennessey[5], Burhan Lehri[7], Chun Ting Angus Lam[1], Priscilla F. Gerber[2], Patricia Lynne Conway[6], Richard A. Stabler[7], Damer Blake[5], Fiona Tomley[5], Dirk Pfeiffer[1,2], Ngoc Thi Pham[3], Guillaume Fournié[5,8,9], Anne Conan[1,10,11]*

**1** Centre for Applied One Health Research and Policy Advice (OHRP), City University of Hong Kong, Hong Kong, SAR China, **2** Department of Infectious Diseases and Public Health, Jockey Club College of Veterinary Medicine and Life Sciences, City University of Hong Kong, Hong Kong, SAR China, **3** National Institute of Veterinary Research, Ha Noi, Vietnam, **4** CIRAD (French Agricultural Research Centre for International Development), Ha Noi, Vietnam, **5** Department of Pathobiology and Population Sciences, Royal Veterinary College, London, United Kingdom, **6** Singapore Centre for Environmental Life Science Engineering (SCELSE), Nanyang Technological University, Singapore, **7** Department of Infection Biology (DIB), London School of Hygiene and Tropical Medicine (LSHTM), London, United Kingdom, **8** Université de Lyon, INRAE, VetAgro Sup, UMR EPIA, Clermont-Auvergne-Rhône-Alpes, Lyon, France, **9** Université Clermont Auvergne, INRAE, VetAgro Sup, UMR EPIA, Clermont-Auvergne-Rhône-Alpes, Clermont-Ferrand, France, **10** French Agricultural Research Centre for International Development (CIRAD), UMR ASTRE, Harare, Zimbabwe, **11** Animals, Health, Territories, Risks, Ecosystems (ASTRE), Univ Montpellier, CIRAD, INRAE, Montpellier, France

* milaxin2-c@my.cityu.edu.hk (CX); anne.conan@cirad.fr (AC)

## Abstract

*Campylobacter* and non-typhoidal *Salmonella* (NTS) are among the most common foodborne pathogens found in chickens at any production stage and cause gastro-enteritis in humans. This study aimed to estimate the prevalence of *Campylobacter* spp. (*C. coli* and *C. jejuni*) and NTS in broiler production and distribution networks (PDNs) using a Bayesian approach. It also investigated the NTS serotypes circulating in these PDNs. A cross-sectional study was conducted in four provinces in northern Vietnam between March 2021 and March 2022. A total of 102 sites, including live bird markets, slaughter facilities (slaughterhouses and slaughter points), and their supplying farms, were randomly selected for sampling. Cecal and environmental samples were cultured for isolation of *Campylobacter* and NTS, with serotypes of NTS determined by targeted analysis of whole genome sequences. Bayesian models were developed to estimate the prevalence of *Campylobacter* at two levels (bird-level and site-level) and NTS at site-level. The selected best-fitted models indicated that *C. jejuni* prevalence was primarily influenced by site type, while *C. coli* was affected by both province and site types. For NTS, only site type was included. The highest overall prevalence of infected broilers was estimated on farms for *C. coli* (26.2%

**Data availability statement:** All relevant data supporting the findings are included in the manuscript and S1 Data file.

**Funding:** This study was funded by the UKRI GCRF One Health Poultry Hub (Grant No. BB/S011269/1), one of twelve interdisciplinary research hubs funded under the UK government's Grand Challenge Research Fund Interdisciplinary Research Hub initiative, awarded to a consortium led by FT and including GF, DB, DP, NPT, RS and PLC. The funders had no role in study design, data collection and analysis, decision to publish, or preparation of the manuscript.

**Competing interests:** The authors have declared that no competing interests exist.

[95% High Density Interval (HDI): 19.0-36.0%]) and *C. jejuni* (19.9% [95%HDI 13.0-27.0%]). Slaughter points (97.6% [95%HDI 63.3-99.9%]) and wholesale markets (91.7% [95%HDI 28.2-99.9%]) had the highest probability of *C. coli* and *C. jejuni* contamination, respectively, but retail markets had the highest proportion of infected broilers at contaminated sites. NTS contamination was more frequent in markets and slaughter facilities (42.8% [95%HDI 30.8-57.1%]) than on farms (18.6% [95%HDI 9.5-30.1%]). Among 16 detected NTS serotypes, *S.* Infantis and *S.* Kentucky were the most common. These findings highlight the widespread contamination of broiler PDNs with *Campylobacter* and NTS in northern Vietnam, emphasizing the need for enhanced surveillance and control measures in PDNs to mitigate the risk of food-borne transmission.

---

## Author summary

*Campylobacter* spp. and non-typhoidal *Salmonella* (NTS) are enteric pathogens that are key global pathogens causing diarrheal diseases. The *Campylobacter* species that are most frequently responsible for human diseases (campylobacteriosis) are *Campylobacter jejuni* (*C. jejuni*) and *Campylobacter coli (C. coli)*. The severity of the salmonellosis depends on host factors and the serotype of NTS. Poultry and poultry products are recognized as major sources of NTS and *Campylobacter* infections in humans. In Vietnam, these pathogens have been detected in the broiler production and distribution networks (PDN). Factors that may increase the risk of transmission of those pathogens include low biosecurity practices, inadequate waste management, and high farm density. In this work, we estimated the prevalence of *Campylobacter* spp. (*C. coli* and *C. jejuni*) and NTS in PDN markets and slaughterhouses and farms supplying these facilities in four provinces in northern Vietnam. We estimated a high prevalence of *Campylobacter* contamination in slaughter points and wholesale markets and higher NTS contamination in slaughter facilities and markets than in farms. *S.* Infantis was one of the most frequently identified serovars. However, it was not reported as being frequently detected in humans in Vietnam. Therefore, serovar may vary depending on the source and region of contamination. Our results help us look at levels of infection/contamination with *Campylobacter* and NTS in the broiler PDNs, thus informing stakeholders to prevent and control campylobacteriosis and salmonellosis.

## Introduction

*Campylobacter* spp. and non-typhoidal *Salmonella* (NTS) are major bacterial pathogens causing intestinal infections and gastroenteritis in humans, accounting for an estimated 80,000 deaths globally in 2010 [1]. Food-producing animals, particularly poultry, serve as asymptomatic reservoirs for these pathogens [2]. *Campylobacter*

spp. can colonize and persist in the intestines of apparently healthy birds [3]. *Campylobacter jejuni* (*C. jejuni*) is the most frequently isolated species in human bacterial foodborne illness, accounting for around 90% of reported infections, while *Campylobacter coli* (*C. coli*) is responsible for most of the remainder [4,5]. Similarly, NTS, which refers to *Salmonella enterica* serovars other than *S.* Typhi and *S.* Paratyphi [6], is frequently detected in poultry gastrointestinal tracts [2,7]. Poultry and poultry products are recognized as major sources of NTS and *Campylobacter* infections in humans [8]: broiler meat is considered the main food-borne source of human campylobacteriosis [9], while *S.* Enteritidis, the NTS serovar most frequently associated with salmonellosis, is primarily linked to consumption of contaminated eggs and poultry meat [9,10]. Additionally, human NTS infections may also be acquired through contaminated environments and water [11].

In Vietnam, poultry production has undergone rapid intensification over the last two decades [12], leading to the emergence of heterogenous production and distribution networks (PDNs) [13,14], in which poultry are bred, raised, traded, and consumed. Most of Vietnam's broiler chicken meat production is sourced from slow-growing broilers, also known as colored broilers, which account for an estimated 72% of the total broiler population [15]. While fast-growing industrial broiler breeds are typically raised for 30–45 days, slow-growing broilers are hybrids of industrial and local breeds that are raised for 70 days to several months within diverse farming systems. Their distribution to consumers involves multiple intermediaries operating at a variety of sites, from farms to PDN endpoints, with the latter referring to sites where chickens are slaughtered or last traded alive (e.g., slaughter facilities, and live bird markets) [15]. These factors, compounded by low biosecurity practices, inadequate waste management, and high farm density, may promote the transmission of pathogens, including *Campylobacter* and NTS [16–19]. Understanding how the prevalence of *Campylobacter* and NTS varies along the PDN is essential to inform the development of targeted surveillance programs and effective risk mitigation interventions.

In Vietnam, a previous study conducted in the Mekong Delta region (southern Vietnam) estimated a prevalence of *C. jejuni* (28.4%) in farmed chickens [20]. In Ha Noi, a high proportion (74%) of retail chicken meat samples were found to be contaminated with *Campylobacter*, predominantly *C. coli* [21]. Similarly, NTS was also highly prevalent (45.9%) in chicken carcasses from retail markets in Vietnam [22]. Although *Campylobacter* and NTS have been detected at multiple stages of Vietnamese poultry PDNs [23–25], there is a lack of studies that assess pathogen prevalences in the diverse types of endpoints and their supplying farms.

Therefore, our objective was to estimate the prevalence of *Campylobacter* spp. (*C. coli* and *C. jejuni*) and NTS in farms and at the PDN endpoints they supply in four provinces in northern Vietnam. NTS serotypes circulating in these PDNs were also identified.

## Materials and methods

### Ethic statement

This study was approved by the National Institute of Veterinary Research (Vietnam) (020–433/DD-YTCC) and the Royal Veterinary College (UK) Ethics and Welfare Committee (URN: 2020 1983–3). All participants informedly agreed to participate in the study. Oral consent was obtained from the farmer to sample their chickens.

### Study area and study design

A cross-sectional study was conducted in four provinces in northern Vietnam between March 2021 and March 2022. Northern Vietnam accounted for 36.5% of the country's human population in 2020 [26] and around 67.3% of the poultry population [12,27]. Ha Noi, Bac Giang, and Hai Duong are among the most densely populated provinces both in terms of humans and poultry in northern Vietnam [26,27], while Quang Ninh is a bordering province with China (Fig 1).

The project aimed to assess the transmission of pathogens between farms and PDN endpoints; the sample size needed to be defined based on the expected size of targeted bacterial populations, which were unknown. Number of sites

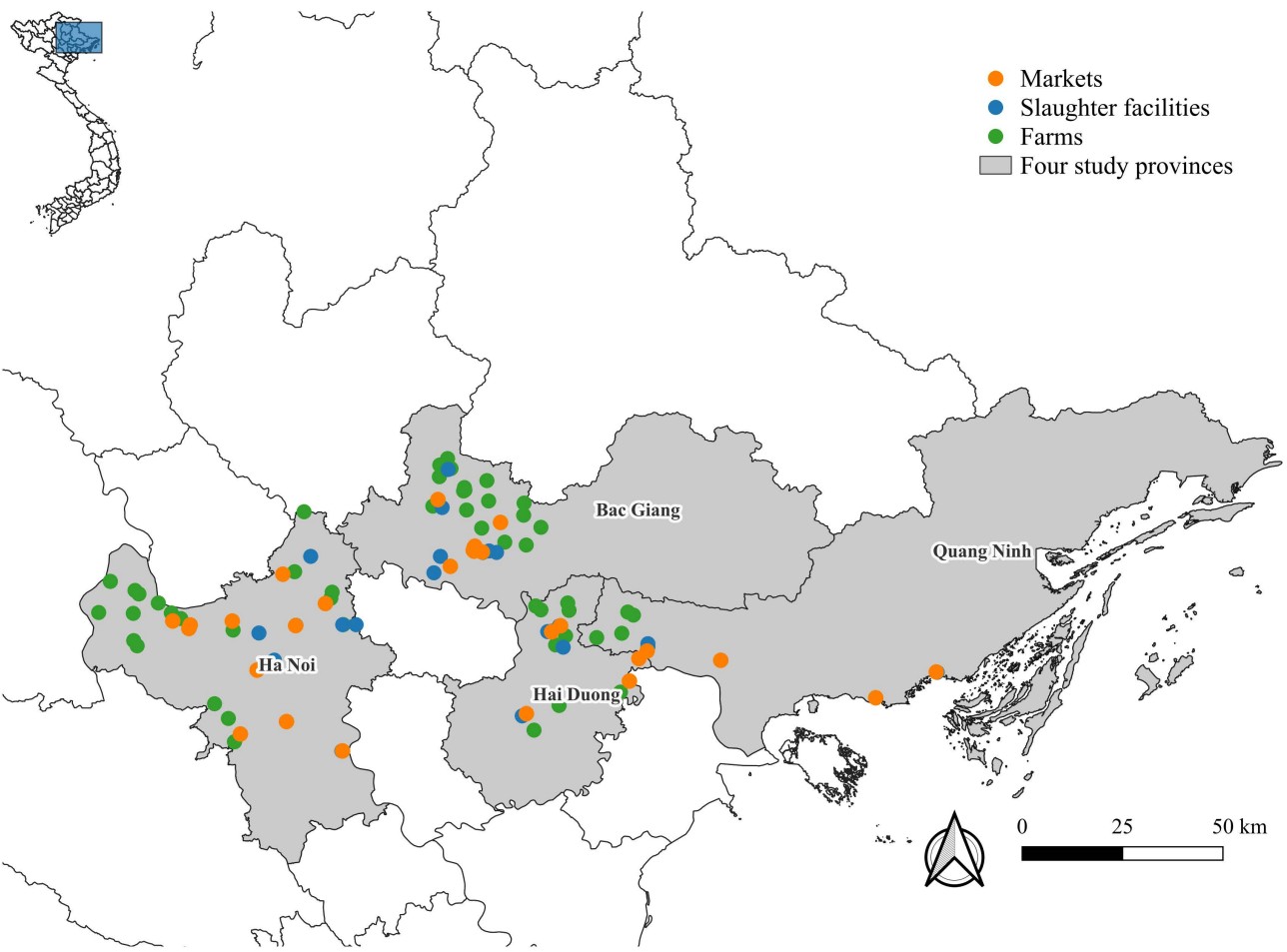

**Fig 1. Location of selected chicken farms (green points), slaughter facilities (blue points), and markets (orange points) in the four study provinces in Vietnam (base layer map source [28]).**

to select was set to sample chickens as widely as possible. Fast- and slow-growing broilers were sampled on 50 farms and 52 supplied PDN endpoints (S1 Table), which included markets and slaughter facilities. Markets were defined as authorized open spaces where at least two vendors operate at least once a week. They were classified as 'wholesale' if some vendors sold chickens to other traders who resold them elsewhere, and 'retail' if all chickens were sold directly to consumers and/or restaurants and caterers. Slaughter facilities were classified as slaughterhouses and slaughter points. Slaughterhouses have automated or partially automated processing chains capable of handling large numbers of chickens. Slaughter points involve manually preparing dozens to hundreds of chickens daily with or without plucking machines.

## Site selection

For endpoints' selection, a stratified cluster sampling design was adopted, with a stratification by provinces. First inclusion criterion was set as endpoints supplying urban areas. Retail markets and slaughter points fitting this criterion were located in urban areas or around urban centers (live bird markets are prohibited in Ha Noi's urban center), while wholesale markets and slaughterhouses were located in peri-urban or rural areas. Meanwhile, slaughter points needed to process at least 30 chickens a day to ensure access to five chickens to sample. In the four provinces, lists of eligible endpoints

(markets and slaughter facilities located in urban and peri-urban areas) were initially compiled through consultation with province and district-level Department of Animal Health officers, and stakeholder interviews in visited markets and slaughter facilities. Finally, due to their small numbers, all eligible wholesale markets (N = 11) and slaughterhouses (N = 6) that processed slow-growing broiler chickens in the four study provinces were recruited. Retail markets (N = 16) and slaughter points (N = 19) were selected through stratified cluster sampling: four to five wards (i.e., urban sub-districts) were randomly selected from each province. In each ward, one retail and one slaughter point were chosen based on the presence of slow-growing broilers. Fast-growing broilers were sampled when available on the site (S1 Table).

Endpoint supply areas, i.e., the provinces where supplying farms were located, were identified through structured interviews of vendors and slaughter facility managers. All farms located within the supply areas and raising at least 100 slow-growing broilers to a late production cycle stage (i.e., ≥ 70 days old) were eligible for inclusion and listed with the assistance of district-level Department of Animal Health officers. To select a paired supplying farm for each endpoint, one commune (i.e., peri-urban or rural sub-district) within the supply areas was randomly selected, and then, a single eligible farm was randomly chosen in that commune.

## Chicken and environment sampling

All selected sites were visited, and oral consent to participate was obtained from the owner or manager of each site. GPS coordinates of sites were recorded, and standardized structured questionnaires were utilized. At each farm and slaughter facility, five slow-growing broilers were randomly selected and purchased for terminal sampling (sampling after euthanasia). In markets, five stalls were randomly selected, and one slow-growing broiler was purchased from each for terminal sampling. When available, five fast-growing broiler chickens were sampled as well. Cecal contents were collected from chicken carcasses and stored at 4°C in a cool box. Environmental samples were collected from each farm using boot socks or overshoes in the broilers' living areas. At each endpoint, samples were obtained by collecting fecal material from the bottom of cages or enclosures where terminally sampled chickens were kept, using an inverted plastic food bag to grab a handful of fecal material from each area. After sampling, each sock, overshoe, or bag was placed into a plastic bag or sampling bottle containing 140 ml of brain heart infusion broth and stored upright in a cool box for further processing. All samples were transported and then processed immediately upon arrival at the laboratory of the National Institute of Veterinary Research (NIVR). The time from collection to processing was within 24 hours.

## Isolation and identification of *Campylobacter spp.* and NTS

Cecal content samples were processed following standard protocols for *Campylobacter* isolation and identification (ISO 10272–1: 2017). Briefly, one loop (10 μl volume) of cecal content was directly placed into Preston agar (Oxoid, UK) and modified charcoal-cefoperazone-deoxycholate agar (mCCDA, Oxoid, UK), then incubated for 44 ± 4 h at 41.5 ± 1 °C in a gas jar with microaerophilic conditions using CampyGen (Oxoid, UK). Five colonies from these plates were screened by Gram-stain and wet mount for typical *Campylobacter* morphology and motility. Typical colonies were subjected to physiological and polymerase chain reaction (PCR) confirmation. For physiological confirmation, after sub-culturing isolates on Columbia sheep blood agar (Oxoid, UK), oxidase, catalase, and hippurate tests were applied [29]. DNA extracts prepared from fresh plate cultures using the Gene JET Genomic DNA Purification Kit (Thermo, USA). Genus and species identification was verified by PCR using primers listed in Table 1, as previously described [30].

For NTS, environmental samples were pre-enriched into brain-heart infusion broth (BHI, Merck, Germany) by incubation at 37°C for 16–18 h. Then, samples were selectively enriched in modified semisolid Rappaport Vassiliadis (MSRV, Merck, Germany) agar with incubation at 41.5°C for 24 to 48h. Positive cultures were streaked onto plates of xylose lysine deoxycholate (XLD, Merck, Germany) agar and incubated at 37°C for 24h. The isolates that showed growth with black or center-black coloring were presumptively identified as *Salmonella* spp. Genomic DNA was extracted using QIAamp DNA Mini Kit (Qiagen, Germany), following the manufacturer's protocol. DNA concentration was quantified using the Qubit

**Table 1. Primers used to identify *Salmonella* and *Campylobacter* genus, *C. jejuni*, and *C. coli* [30].**

| Name of Primer | Identification level | Sequence (5'-3') | Target | Product Size (bp) |
|---|---|---|---|---|
| 23S rRNA F | Genus | TATACCGGTAAGGAGTGCTGGAG | *C. jejuni 23S rRNA* | 650 |
| 23S rRNA R | | ATCAATTAACCTTCGAGCACCG | | |
| CJF | Species | ACTTCTTTATTGCTTGCTGC | *C. jejuni hipO* | 323 |
| CJR | | GCCACAACAAGTAAAGAAGC | | |
| CCF | Species | GTAAAACCAAAGCTTATCGTG | *C. coli glyA* | 126 |
| CCR | | TCCAGCAATGTGTGCAATG | | |
| ST11 | Genus | AGCCAACCATTGCTAAATTGGCGCA | *Salmonella* spp. | 429 |
| ST15 | | GGTAGAAATTCCCAGCGGGTACTG | | |

dsDNA High Sensitivity (HS) Assay Kit (Thermo Fisher Scientific) prior to library preparation. Confirmation of NTS identity was performed using PCR with primers ST11 and ST15 (Table 1).

## Serotyping of the isolated NTS

Library preparation was performed according to the Illumina TruSeq Nano DNA sample preparation protocol as previously described [31–33]. In brief, DNA samples were sheared to ~450 bp using a Covaris E220 ultrasonicator, following the manufacturer's guidelines. Libraries were uniquely tagged with Illumina's IDT for Illumina-TruSeq DNA Unique Dual Indexes to enable multiplexing during sequencing. Finished libraries were quantified using Promega's QuantiFluor dsDNA assay, and the average library size was determined on an Agilent Tapestation 4200 (Agilent, USA). Library concentrations were normalized to 4 nM and validated by qPCR on a QuantStudio-3 real-time PCR system (Applied Biosystems) using the Kapa library quantification kit for Illumina platforms as recommended by the manufacturer (Kapa Biosystems). Equimolar pools of the prepared libraries were sequenced on the Illumina HiSeq X platform at a read length of 150 bp paired-end.

Raw read quality filtering was performed using Cutadapt (version 2.10) to remove adapter sequences and low-quality bases. The filtering parameters were set at an error rate of 0.1 (-e 0.1), a quality cutoff of 20 (-q 20), and a minimum read length of 30 (--minimum-length 30). Trimmed reads were then de novo assembled using SPAdes (version 3.15.3) to generate draft genomes. The assembled genomes were analyzed using tools from the Centre for Genomic Epidemiology (CGE) to characterize various genomic features. Serotyping of *Salmonella* isolates was then determined using SeqSero2. The analytical approach used in this study follows established WGS-based methodologies for foodborne pathogen characterization [31–33].

## Bayesian analysis

**Model description.** Bayesian logistic regression models were developed to estimate the prevalence of *Campylobacter* at two levels (bird-level and site-level) and NTS at site-level. Separate models were specified for *C. coli*, *C. jejuni*, and NTS. Specifically, we developed two-level Bayesian hierarchical logistic regression models for both *C. coli* and *C. jejuni*, treating the site as a random effect. In each model, bird- and site-level factors were integrated to assess the risk of infection in broilers. The contamination status of a site follows a Bernoulli distribution, with the probability of a site being contaminated depending on its type. Then, the probability of a broiler testing positive in a contaminated site was formulated using logistic regression with a site-specific random intercept to account for unobserved heterogeneity at each site and explanatory fixed variables: chicken type, provincial location, and site type. As one environmental sample was collected at each site to test for the presence of NTS, the probability of a site being contaminated with NTS was modeled using logistic regression, with provincial location and site type as explanatory

variables. Chicken type was not included as a variable because some broiler compositions were represented by only a small number of sites, with two sites exclusively housing fast-growing broilers. Site types were differentiated into (i) farm and endpoint, (ii) farm, market, and slaughter facility, or (iii) farm, retail market, wholesale market, slaughter point, and slaughterhouse. Chicken type was distinguished between slow- and fast-growing broilers. For NTS models, the provincial location referred to the location of the sampling site, as the contamination levels at these sites might vary across different provinces. In contrast, for *Campylobacter* models, the provincial location could refer to either the location of the site or the province of the supply area. This distinction is important because the province of the supplying farms could influence the prevalence of *Campylobacter* spp. in broilers, which were transported along different levels of the PDNs. The province of the supply area was not included in the NTS model due to the limited number of observations. Site locations were Ha Noi, Bac Giang, and Hai Duong/Quang Ninh, as most sites in Hai Duong and Quang Ninh were located close to their shared border (Fig 1), and supply areas were Ha Noi, Bac Giang, Hai Duong/Quang Ninh, and other provinces. Note that each site supply area was a dichotomous explanatory variable as a given site could be supplied by several of these areas. Weakly informative priors were specified: *Beta*(1,1) for the probability of a site being contaminated, *Normal*(0, 4) for the fixed regression coefficients, and an *InverseGamma*(1, 1) for the variance of the random intercept.

## Model development

Models with all combinations of explanatory variables were fitted (N = 12 each for *C. coli* and *C. jejuni* models, and N = 5 for the NTS model). Provincial location was only considered in models for which all site types (farm, retail market, wholesale market, slaughterhouse, and slaughter point) were accounted for, as these types of endpoints and farms were heterogeneously distributed across the four provinces. Models were run using Markov Chain Monte Carlo simulation (MCMC) in JAGS (Just Another Gibbs Sampler) [34] and R.4.4.2 [35]. For each model, we ran four separate chains with an initial burn-in period of 10,000. The Deviance Information Criterion (DIC) was computed for each model. A model with a lower DIC value was considered a better fit and better supported than others if the difference in DIC scores was at least five units (ΔDIC ≥ 5). Convergence and mixing within the Markov chain were assessed by examining trace plots. All model effective sample sizes were also higher than 1000, given that a random intercept was considered, which is sufficient for stable estimates [36], and the Gelman-Rubin convergency statistics were rounded to or less than 1.01, with successful convergence based on Rhat values (all < 1.1) [37]. Pair plots of the posterior distributions were generated, enabling us to assess parameter correlations and understand the distribution of each parameter's posterior values. Moreover, posterior predictive checks (PPCs) were created to assess the capability of the models to replicate observed data patterns. Finally, we conducted Bayesian retrospective power analyses for *Campylobacter* and NTS models to evaluate the models' ability to detect differences of 10% in prevalence and contamination probability between the different explanatory variables. The HDI of the estimated power was also computed [38].

## Results

### Descriptive results for *C. coli* and *C. jejuni*

A total of 500 slow-growing broilers and 65 fast-growing broilers were sampled in 102 sites from four provinces. Among these, 146 (25.8%) tested positive for *C. coli* and 101 (17.9%) for *C. jejuni* (Table 2). *Campylobacter coli* was most prevalent in slaughterhouses (n = 18, 45.0%), while C. *jejuni* was most prevalent on farms (n = 56, 22.4%). Conversely, wholesale markets had the lowest proportion of broilers positive for *C. coli* (n = 8, 11.4%), and no broilers from slaughterhouses tested positive for *C. jejuni* (n = 0, 0%).

Notably, the site location associated with the slightly highest proportion of positive broilers was Ha Noi province for *C. coli* (n = 67, 29.1%) and Hai Duong/Quang Ninh provinces for *C. jejuni* (n = 32, 19.4%), as shown in Table 3.

**Table 2. The number and proportions of chicken samples testing positive by chicken types and by site types for *Campylobacter* spp. (observed data).**

| | | *Campylobacter* spp. | | | |
| | | *C. coli* | | *C. jejuni* | |
| Variables | | Positive samples [n, (%)] | no. of samples | Positive samples [n, (%)] | no. of samples |
|---|---|---|---|---|---|
| Chicken Type | Slow-growing broiler | 132 (26.4) | 500 | 96 (19.2) | 500 |
| | Fast-growing broiler | 14 (21.5) | 65 | 5 (7.7) | 65 |
| | Total | 146 (25.8) | 565 | 101 (17.9) | 565 |
| Site Type | Farm | 73 (29.2) | 250 | 56 (22.4) | 250 |
| | Retail | 16 (20.0) | 80 | 15 (18.8) | 80 |
| | Wholesale | 8 (11.4) | 70 | 9 (12.9) | 70 |
| | Slaughterhouse | 18 (45.0) | 40 | 0 (0.0) | 40 |
| | Slaughter point | 31 (24.8) | 125 | 21 (16.8) | 125 |
| | Total | 146 (25.8) | 565 | 101 (17.9) | 565 |

**Table 3. The number and proportion of chicken samples testing positive by site provinces for *Campylobacter* spp. (observed data).**

| Bacteria | Site province | Positive samples [n/N, (%)]* | Positive samples by broiler type | | Positive samples by site type | | | | |
| | | | Slow-growing broiler [n, (%)] | Fast-growing broiler [n, (%)] | Farm [n, (%)] | Retail [n, (%)] | Wholesale [n, (%)] | Slaughter-house [n, (%)] | Slaughter point [n, (%)] |
|---|---|---|---|---|---|---|---|---|---|
| *C. coli* | Bac Giang | 42/170 (24.7) | 37 (21.8) | 5 (2.9) | 20 (11.8) | 6 (3.5) | 4 (2.4) | 6 (3.5) | 6 (3.5) |
| | Ha Noi | 67/230 (29.1) | 60 (26.1) | 7 (3.0) | 34 (14.8) | 4 (1.7) | 4 (1.7) | 12 (5.2) | 13 (5.7) |
| | Hai Duong/ Quang Ninh | 37/165 (22.4) | 35 (21.2) | 2 (1.25) | 19 (11.5) | 6 (3.6) | 0 (0) | 0 (0) | 12 (7.3) |
| *C. jejuni* | Bac Giang | 30/170 (17.6) | 30 (17.6) | 0 (0) | 19 (11.2) | 1 (0.6) | 0 (0) | 0 (0) | 10 (5.9) |
| | Ha Noi | 39/230 (17.0) | 36 (15.7) | 3 (1.3) | 20 (8.7) | 5 (2.2) | 9 (3.9) | 0 (0) | 5 (2.2) |
| | Hai Duong/ Quang Ninh | 32/165 (19.4) | 30 (18.2) | 2 (1.2) | 17 (10.3) | 9 (5.5) | 0 (0) | 0 (0) | 6 (3.6) |

*n = number of positive chicken samples; N = total number of chicken samples tested; % = percentage of positive samples

## Descriptive results and serotypes of NTS

A total of 102 environmental samples were collected at 102 sites (Table 4), with 32 (31.4%) testing positive for NTS, representing 16 distinct serotypes (Table 5). The highest occurrence of NTS was in environmental samples collected from wholesale markets (n = 6, 54.5%), and the lowest in farm samples (n = 9, 18%). The most frequently detected serotypes were *S.* Infantis and *S.* Kentucky, with *S.* Infantis identified in two markets and two slaughter facilities and *S.* Kentucky in two farms and two slaughter facilities. *Salmonella* Agona, *S.* Albany/Duesseldorf, and *S.* London were each detected in three sites, and other serovars were detected at either two sites or just one site.

## Bayesian model results of *C. coli* and *C. jejuni*

**Best models and post-predictive checks.** Twelve models were specified for *C. coli* and *C. jejuni* (Table 6). Convergence was successfully achieved for all models, with Gelman-Rubin convergency statistics rounded to or below 1.01. The trace (n chains = 4, n iterations = 50000) of these two selected models (one for *C. coli* and one for *C. jejuni*) displayed tight, consistent horizontal bands, indicating no visual signs of non-convergence within the chains

**Table 4. The number and percentage of environmental samples testing positive in different sites for NTS (observed data).**

| Site type | Positive samples [n, (%)] | Total no. of samples |
|---|---|---|
| Farm | 9 (18.0%) | 50 |
| Retail | 6 (37.5%) | 16 |
| Wholesale | 6 (54.5%) | 11 |
| Slaughterhouse | 3 (50.0%) | 6 |
| Slaughter point | 8 (42.1%) | 19 |
| Total | 32 (31.4%) | 102 |

**Table 5. The serotype of NTS identified by site type.**

| Predicted serotype | No. of serotypes in different site types | | | | | |
| | Farm | Retail | Wholesale | Slaughterhouse | Slaughter point | Overall |
|---|---|---|---|---|---|---|
| Infantis | 0 | 0 | 2 | 1 | 1 | 4 |
| Kentucky | 2 | 0 | 0 | 1 | 1 | 4 |
| Agona | 0 | 1 | 1 | 0 | 1 | 3 |
| Albany or Duesseldorf | 1 | 1 | 0 | 0 | 1 | 3 |
| London | 1 | 2 | 0 | 0 | 0 | 3 |
| Brancaster | 1 | 0 | 0 | 0 | 1 | 2 |
| Enteritidis | 0 | 0 | 1 | 0 | 1 | 2 |
| I 4, [5],12:i:- | 0 | 1 | 0 | 1 | 0 | 2 |
| Indiana | 1 | 0 | 1 | 0 | 0 | 2 |
| Newport | 0 | 0 | 1 | 1 | 0 | 2 |
| Weltevreden | 2 | 0 | 0 | 0 | 0 | 2 |
| Corvallis | 0 | 0 | 0 | 0 | 1 | 1 |
| I 4:b:- | 1 | 0 | 0 | 0 | 0 | 1 |
| Rissen | 0 | 1 | 0 | 0 | 0 | 1 |
| Schwarzengrund | 1 | 0 | 0 | 0 | 0 | 1 |
| Typhimurium | 0 | 0 | 0 | 0 | 1 | 1 |

(S1 and S2 Figs). Pairs plots for parameters of two selected *C. coli* and *C. jejuni* models sampled via the MCMC algorithm from four chains are shown in S3 and S4 Figs. The selected *C. coli* model differentiated for all site types (farm, retail, wholesale, slaughterhouse, and slaughter point) and accounted for supply areas (Ha Noi, Bac Giang, Hai Duong/Quang Ninh, and other provinces). This model was 4.2 DIC units lower than the model that used site provincial locations instead of supply areas while maintaining similar site-type coefficient estimates. A model that included chicken type had a slightly lower DIC ($\Delta$DIC = 0.7), but this additional parameter value was not significantly different from zero, and other parameter estimates remained consistent with the selected model. The posterior predictive checks of the selected model (S5 Fig) revealed a notable discrepancy for slaughterhouses, where the observed proportion of chickens positive for *C. coli* (42.6%) far exceeded the model's predicted median (9.8% [95%High Density Interval (HDI): 2.5-31.6%]). For *C. jejuni* models, the models differentiating all site types achieved DIC scores at least 12.7 DIC units lower than any other models with a different site-type specification. The selected model (Table 6) also excluded chicken type and provincial location, as the inclusion of these variables did not substantially improve the model fit, and did not alter site-level parameter estimates. The PPCs of the selected *C. jejuni* model show that posterior predictions for wholesale markets underestimated the observed proportion of *C. jejuni*-positive broilers (predicted: 4.1% [95% HDI 1.0–13.3%]; observed: 13.2%) (S6 Fig).

PLOS Neglected Tropical Diseases

**Table 6. The summary of DIC values for twelve models for *C. coli* and *C. jejuni* separately. The selected model based on best fit with DIC and parameters evaluation is highlighted with a star.**

| Models for *C. coli* | | | | Models for *C. jejuni* | | | |
|---|---|---|---|---|---|---|---|
| Site Type | Chicken type | Province | DIC | Site Type | Chicken type | Province | DIC |
| 1 | 1 | 1 | 396.3 | 1 | 1 | 1 | 331.1 |
| 1 | 2 | 1 | 397.9 | 1 | 2 | 1 | 331.0 |
| 2 | 1 | 1 | 360.2 | 2 | 1 | 1 | 320.0 |
| 2 | 2 | 1 | 372.5 | 2 | 2 | 1 | 318.3 |
| 3 | 1 | 1 | 363.0 | 3 | 1 | 1 | 307.3 |
| 3 | 2 | 1 | 365.1 | 3 | 2 | 1 | 307.8 |
| 5 | 1 | 1 | 355.1 | **5 ★** | **1** | **1** | **292.3** |
| 5 | 1 | 3 | 349.5 | 5 | 1 | 3 | 292.3 |
| **5★** | **1** | **supply** | **345.3** | 5 | 1 | supply | 294.4 |
| 5 | 2 | 1 | 355.8 | 5 | 2 | 1 | 294.3 |
| 5 | 2 | 3 | 349.9 | 5 | 2 | 3 | 295.1 |
| 5 | 2 | supply | 344.6 | 5 | 2 | supply | 295.1 |

Note: Chicken type was added as 1) any chicken type (not treated as a specific influencing factor) and 2) slow or fast-growing broiler. Province was organized into three classifications: 1) any province (not treated as a specific influencing factor), 3) site province (Ha Noi, Bac Giang, Hai Duong, and Quang Ninh, or 4) province of supply area (Ha Noi, Bac Giang, Hai Duong/Quang Ninh, or other provinces). Meanwhile, site types were tested both at the bird-level and site-level, into 1) any type (not treated as a specific influencing factor), 2) farm or endpoint, 3) farm, markets, or slaughter facilities, and 5) farm, retail, wholesale, slaughter point, or slaughterhouse.

## Posterior prevalence estimate

The estimated prevalences are reported in Tables 7 and 8. *C. coli* had the highest overall prevalence of positive broilers on farms (Mode 26.2% [95% HDI 19.0-36.0%]), while *C. jejuni* showed the highest prevalence on farms (Mode 19.9% [95% HDI 13.0-27.0%]) and in retail markets (Mode 19.8% [95% HDI 9.0-34.0%]). For *C. jejuni*, slaughter points and wholesale markets had the highest levels of site contamination, but the proportion of positive broilers at these contaminated sites was much lower than on farms and in retail markets, where nearly one in three broilers tested positive when the site was contaminated. For *C. coli*, nearly all farms were expected to be contaminated, and the proportion of positive broilers at contaminated sites showed less variability than for *C. jejuni*, with retail markets having the highest prevalence, comparable to that of *C. jejuni*. Both *C. coli* and *C. jejuni* prevalences were substantially lower in slaughterhouses

**Table 7. Probability of site-level and bird-level *C. coli* occurrence in different sites and chicken types (mode; odds ratio [OR], plus 95% High-Density Interval (HDI)) (model-based posterior estimates).**

| Site type | Site-level | Bird-level in *C. coli* contaminated sites | | Bird-level in any sites |
|---|---|---|---|---|
| | *C. coli* mode prevalence, % (95% HDI) | *C. coli* mode prevalence, % (95% HDI) | OR (95% HDI) | *C. coli* mode prevalence, % (95% HDI) |
| Farm | 97.1 (80.0-99.9) | 28.7 (20.8-40.1) | Ref. | 26.2 (19.0-36.0) |
| Retail | 63.3 (36.6-91.2) | 35.8 (16.6-55.1) | 1.3 (0.3-3.1) | 21.6 (10.0-36.0) |
| Wholesale | 86.8 (50.6-99.9) | 28.2 (13.0-52.8) | 1.1 (0.2-2.7) | 22.5 (9.0-42.0) |
| Slaughterhouse | 43.6 (12.9-94.6) | 25.7 (4.9-67.3) | 1.1 (0.04-5.2) | 10.6 (1.0-34.0) |
| Slaughter point | 97.6 (63.3-99.9) | 19.8 (10.4-34.2) | 0.6 (0.2-1.2) | 17.1 (9.0-28.0) |

**Table 8. Probability of site-level and bird-level of *C. jejuni* occurrence in different sites and chicken types (mode; odds ratio [OR], plus 95% High Density Interval (HDI)) (model-based posterior estimates).**

| Site type | Site-level | Bird-level in *C. jejuni* contaminated sites | | Bird-level in any sites |
|---|---|---|---|---|
| | *C. jejuni* mode prevalence, % (95% HDI) | *C. jejuni* mode prevalence, % (95% HDI) | OR (95% HDI) | *C. jejuni* mode prevalence, % (95% HDI) |
| Farm | 69.8 (51.1-92.7) | 28.1 (17.0-40.7) | Ref. | 19.9 (13.0-27.0) |
| Retail | 57.6 (30.1-86.1) | 37.7 (16.9-57.5) | 1.5 (0.3-3.5) | 19.8 (9.0-34.0) |
| Wholesale | 91.7 (28.2-99.9) | 4.1 (0.6-19.3) | 0.2 (0.01-0.6) | 2.9 (0-11.0) |
| Slaughterhouse | 30.9 (8.3-97.8) | 2.9 (0.1-42.9) | 0.2 (0.001-1.9) | 1.3 (0-15.0) |
| Slaughter point | 87.7 (52-99.9) | 16.3 (7.7-30.4) | 0.5 (0.2-1.2) | 12.6 (6.0-22.0) |

compared to other endpoint types. For *C. coli*, having broilers supplied from Ha Noi (OR 1.3 [95% HDI 0.4-2.8]) and Hai Duong/Quang Ninh (OR 1.4 [95% HDI 0.4-3.1]) increased the likelihood of broilers testing positive more than supplies from Bac Giang (OR 0.5, [95% HDI 0.2-1.1]) and other provinces (OR 0.7, [95% HDI 0.2-1.6]).

The ability of the model to detect meaningful differences in *Campylobacter* prevalence was estimated to have a power of 68% (95% HDI 59–77%) for farm vs endpoints at the site level, 35% (95% HDI 26–45%) at the bird level, 28% (95% HDI 20–75%) for slow-growing vs. fast-growing broiler and 61% (95% HDI 51–70%) for province.

## Bayesian model results of NTS

A total of five models were examined for NTS (Table 9). Convergence was successfully achieved for all models, with Gelman-Rubin convergency statistics below 1.01. The trace (n chains = 4, n iterations = 50000) showing consistent horizontal bands, and pairs plots for parameters of the selected NTS model are shown in S7 and S8 Figs. The selected NTS model (Table 9, highlighted with a star symbol) differentiated between farms and endpoints and had 7.6 DIC units lower than a model that included the provincial location. A model that categorized sites into three types (farm, market, and slaughter facility) had a slightly higher DIC score (ΔDIC = 1.3) and did not improve model fit, as the posterior predicted prevalence in markets and slaughter facilities remained similar and was equivalent to the predicted prevalence for endpoints in the two-category model. In this selected NTS model, no discrepancy was observed in the post-predictive check of the Bayesian model for NTS infection (S9 Fig).

**Table 9. The summary of DIC values for ten models for non-typhoidal *Salmonella*. The selected model based on best fit with DIC and parameters evaluation is highlighted with a star.**

| Site type | Province | DIC |
|---|---|---|
| 1 | 1 | 128.9 |
| 2 ★ | 1 | 122.4 |
| 3 | 1 | 124.4 |
| 5 | 1 | 127.5 |
| 5 | 3 | 130.0 |

Note: We used different site types (1) any type (not treated as a specific influencing factor), 2) farm or endpoint, 3) farm, markets, or slaughter facilities, and 5) farm, retail, wholesale, slaughter point, or slaughterhouse), and the province of sites (1) any province (not treated as a specific influencing factor), 3) site province (Ha Noi, Bac Giang, or Hai Duong and Quang Ninh) as risk factors.

**Table 10. The probability of environmental sample testing positive for Nontyphoidal *Salmonella* in farms and endpoint (model-based posterior estimates).**

| Site type | Prevalence of positive environmental samples in sites, mode, %, (95% HDI) | OR (95% HDI) |
|---|---|---|
| Farm | 18.6 (9.5-30.1) | Ref. |
| Endpoint | 42.8 (30.8-57.1) | 3.3 (1.1-7.2) |

The environment of markets and slaughterhouses (Mode 42.8% [95% HDI 30.8-57.1%]) was more likely to be contaminated by NTS than farms (Mode 18.6% [95% HDI 9.5-30.1%]) (Table 10). The estimated power of the NTS model to detect meaningful differences of contamination between farms and endpoints was 79% (95% HDI 70–86%).

## Discussion

Our study shows that prevalence of *Campylobacter* spp. in broilers, and NTS environmental contamination of broiler facilities, are both high in northern Vietnam. While the prevalence and contamination vary between sites, it is noted that both pathogens were detected in all types of sites, namely farms, markets, and slaughter facilities. These results are consistent with previous studies in southern Vietnam [20,39] or Ha Noi's markets [21,40]. For example, in Ha Noi a high prevalence of *Campylobacter* spp. (41.1%) was reported on fresh chicken carcasses at retail markets, with the predominant species being *C. coli*, followed by *C. jejuni* [40]. Another study from 2012 in the Mekong Delta region (southern Vietnam) reported a high prevalence of NTS colonization (45.6%) in 204 backyard chicken farms [39]. In our study, the most commonly identified NTS serotype was *S.* Infantis, one of the *Salmonella enterica* serovars associated with infections in both humans and broilers [41]. It was one of the top five serotypes in NTS human infections in Europe in 2017 [42]. In contrast, *S.* Infantis has not been reported as being frequently detected in humans in Vietnam; the serovar most frequently isolated in non-invasive NTS human disease was *S.* Typhimurium [43], which was not the case in our findings. Similarly, *S.* Weltevreden, the second most detected serovar in humans [43], was not as prevalent in our study either. This may be explained by variations in contamination sources across different regions. For example, *S.* Typhimurium was more common in pork production in Vietnam [44]; this could explain its high detection in humans (high consumption of pork in Vietnam). To investigate these discrepancies, surveillance of the NTS with serotyping of the strain in humans and poultry would be beneficial in identifying the sources of the contamination. Moreover, our overall results raise concerns about the spread of foodborne bacterial pathogens and their impact on food safety.

Notably, we have found the overall prevalence of *Campylobacter* infection in broilers to be generally high in farms, and retail markets, with the highest prevalence estimated to be on farms for *C. jejuni* (19.9%) and *C. coli* (26.2%). The higher prevalence on farms may be attributed to potential contamination from suppliers, such as a high prevalence of *Campylobacter* in the supplied day-old chicks. The typical timing of *C. jejuni* colonization in chickens occurs when they are approximately seven days old [45]. Another possible cause of *Campylobacter* spp. could be low biosecurity in these farms and the proximity of other livestock production. An earlier study in the Mekong Delta of Vietnam reported that it was common for farms to have a combination of animal species, with 35% of pig farms also raising chickens and 18% of poultry farms keeping pigs. In these settings, *C. jejuni* was the most common species detected in chickens (20%) in poultry farms with low biosecurity and mixed-species farming, contributing to its spread [20].

The highest prevalence of positive broilers of *C. jejuni* (37.7%) and *C. coli* (35.8%) in contaminated sites were estimated at retail markets. Similarly, in a study conducted in 2005 in Ha Noi, Vietnam, high contamination rates were observed in retail chicken products, with the most frequently isolated *Campylobacter* being *C. jejuni* (45.2%), followed by *C. coli* (25.8%) in retail fresh chicken carcasses [46]. Broilers at retail markets may have spent longer periods being transported through the PDN compared to other endpoints. For instance, slow-growing broiler chickens in Vietnam are first collected from different farms in multiple locations by mobile traders with the assistance of middlemen, who later sell

these broilers to other traders in local wholesale live bird markets, where broilers are further sold to retailers [15]. This highly connected transport and longer broiler transportation process may promote the transmission of pathogens (*Campylobacter* spp.) within and between different network sites through inadequate sanitized transport vehicles and cages, as well as through traders and intermediaries. Extended transit periods and longer transport distances can also increase transport-induced stress in broilers, enhancing the growth and shedding of *Campylobacter* spp. [47]. Meanwhile, broilers from multiple origins are generally mixed in retail markets. Infected broilers can introduce pathogens into the retail market population and contaminate environments, increasing the risk of cross-contamination. Therefore, it's important to regularly and properly clean and disinfect retail markets, as well as the transport material used for broiler distribution.

Another key finding was the high proportion of *C. coli* (97.6%) and *C. jejuni* (87.7%) contaminated slaughter points. An earlier study from Senegal also found a high prevalence (64.4%) of *C. jejuni* among four *Campylobacter* species in chicken neck-skin samples at points of slaughter, where 95% of the slaughtering was directly conducted by the seller [48]. At slaughter points, live chickens are kept and slaughtered in the same or adjacent areas, making contaminated surfaces a potential source of infection for other live chickens [49]. In contrast, in slaughterhouses, the lairage area is physically separate from the slaughter line, decreasing chances for surface contamination during evisceration, which could prevent cross-contamination and further reduce *Campylobacter* infection in broilers. Moreover, the waiting time at the slaughter point may be longer with a mix of chicken sources (as for retail markets), increasing the contact between contaminated environments, infected birds, and susceptible birds. Conversely, broilers at slaughterhouses are produced on contract farms with stricter biosecurity measures and are sent directly to the companies' slaughterhouses for processing, making their transport time in PDNs shorter than for slaughter points. Therefore, broilers are less likely to get *Campylobacter* infection in slaughterhouses. However, the extremely low prevalence of *C. jejuni* infection may also stem from other factors, such as the small number of slaughterhouses sampled (N = 6). Nevertheless, good slaughtering practices should be emphasized in all slaughter facilities, and the time broilers are held should be minimized.

The proportion of NTS contamination in markets and slaughter facilities (42.8%) was higher than in farms (18.6%). One study conducted between 2014 and 2015 in wet markets in Penang and Perlis, Malaysia, also found that *Salmonella* was highly prevalent (100%) in poultry processing environments in wet markets and small-scale processing plants [50]. One potential explanation for the higher prevalence of NTS contamination in endpoints may be associated with poor hygiene practices and insufficient sanitation at markets and slaughter facilities. These conditions could provide favorable environments that serve as reservoirs for NTS and facilitate their growth and spread. Furthermore, the stress associated with transporting poultry before slaughter has been shown to also increase NTS populations in both the intestinal tract and on the exterior of the carcass [51], ultimately contributing to the environmental contamination of markets and slaughter facilities. Thus, measures must be taken to reduce stress of broilers during transportation.

Our study has some limitations. Firstly, the modeling approaches differed between *Campylobacter* spp. and NTS, primarily because detecting NTS at the individual level is challenging [52]. Therefore, it is not possible to accurately assess the individual prevalence of NTS in broilers, nor to compare *Campylobacter* prevalence and NTS contamination probability. Secondly, our model did not account for seasonal variations (autumn, winter, spring, and summer) as a risk factor. Elevated temperatures in summer create a favorable environment that significantly boosts bacterial growth. Our sampling period lasted 12 months, covering all four seasons from March 2021 to March 2022. However, this seasonal effect may not act as a confounder as all types of sites were sampled during all seasons. Additionally, other farm-level and endpoint-level risk factors were not assessed, given their heterogeneous nature across farms and endpoints. Thirdly, our site selection process may have introduced selection bias because we lacked an exhaustive list of endpoints, particularly retail markets and slaughter points. We addressed this by conducting snowball sampling interviews to identify additional eligible endpoints; however, some bias remain possible. Moreover, we collected samples from only six slaughterhouses and eleven wholesale markets. This limitation stems from the small number of such facilities in the study area. Given the small sample size, our estimation, based on the Bayesian framework with weakly informative priors, might introduce limited bias

in posterior estimates [38] and led to a relatively low power for some variables. The absence of a difference between type of birds may then be due to this low statistical power. However, despite these limitations, the strength of this study lies in the Bayesian approach, which integrates prior knowledge with observed data, as well as using probability distributions instead of single-point estimates to maintain uncertainties and variations in the model parameters and allows for continuous improvement of the model's accuracy through the integration of new data [38]. Also, the observed prevalence and contamination probabilities may be underestimated. False negatives can occur due to failure in amplification during PCR or qPCR assays, even under optimal conditions. We minimized this risk by ensuring short sample transport times and validating PCR protocols rigorously. Finally, this study did not include human health surveillance data to directly link the observed pathogen contamination in chickens with human disease burden in Vietnam. However, given that contaminated chicken products are a known major source of human *Campylobacter* and NTS infections [2], our findings provide valuable information on the prevalence and contamination levels of foodborne pathogens in the chicken food system.

In summary, our study reflects the epidemiological situation of foodborne pathogens in different network sites in PDNs in northern Vietnam, indicating a high prevalence of *Campylobacter* contamination in slaughter points and wholesale markets. We estimate that markets and slaughter facilities were more likely to be contaminated by NTS than farm environments, with a high diversity of NTS serovars across different sites. These findings highlight the need for integrated intervention measures to enhance food safety, particularly in farms, retail markets, and slaughter points. Interventions to control risk factors associated with broiler transportation along the PDNs are needed. Such interventions could include developing and enforcing strict hygiene and sanitation protocols with regulations for markets where broilers from different geographical origins are mixed and could, therefore, be hotspots of disease transmission. Transport vehicles, cages, traders, and intermediaries, which all pose potential fomite risk, should be regularly sanitized and monitored to help prevent the spread of infections. Continuous monitoring of foodborne pathogens and investigation of associated risk factors is crucial to improving poultry production in Vietnam.

## Supporting information

**S1 Table. Number of selected sites and the number of chickens sampled by different site types.**
(DOCX)

**S1 Fig. Trace plots with four chains for the selected ideal *C. coli* model.** This figure shows the trace plots (n chains = 4, n iteration = 50000) of the selected *C. coli* model, which displayed tight, consistent horizontal bands, indicating no visual signs of non-convergence within the chains. The parameters p_site_1, p_site_2, p_site_3, p_site_4, and p_site_5 represent the *C. coli* contamination probabilities for farms, retail markets, wholesale markets, slaughterhouses, and slaughter points, respectively. The parameter intercept_bird represents the baseline probability of a broiler testing positive for *C. coli* at a contaminated site, assuming no effect from site type and province of supply areas. The fixed-effect coefficients beta_ret, beta_slh, beta_slp, and beta_who quantify the effects of retail markets, slaughterhouses, slaughter points, and wholesale markets on the probability of a broiler testing positive for *C. coli* at a contaminated site, respectively. The fixed-effect coefficients beta_cbac, beta_chdqn, beta_chon, and beta_cot quantify the effects of supply areas—Bac Giang, Hai Duong/Quang Ninh, Ha Noi, and other provinces—on the probability of a broiler testing positive for *C. coli* at a contaminated site, respectively. The parameter sigma_1 means the variance of the random intercept.
(TIFF)

**S2 Fig. Trace plots with four chains for the selected ideal *C. jejuni* model.** This figure shows the trace plots (n chains = 4, n iteration = 50000) of the selected *C. jejuni* model, which displayed tight, consistent horizontal bands, indicating no visual signs of non-convergence within the chains. The parameters p_site_1, p_site_2, p_site_3, p_site_4, and p_site_5 represent the *C. jejuni* contamination probabilities for farms, retail markets, wholesale markets, slaughterhouses, and slaughter points, respectively. The parameter intercept_bird represents the baseline probability of a broiler testing

positive for *C. jejuni* at a contaminated site, assuming no effect from site type. The fixed-effect coefficients beta_ret, beta_who, beta_slh, and beta_slp quantify the effects of retail markets, wholesale markets, slaughterhouses, and slaughter points on the probability of a broiler testing positive for *C. jejuni* at a contaminated site, respectively. The parameter sigma_1 means the variance of the random intercept.
(TIFF)

**S3 Fig. Pairs plot with four chains for the selected ideal *C. coli* model.** This figure shows the pairs plot of the selected *C. coli* model, which showed the distribution of each parameter's posterior values and its interactions with other parameters. The parameters p_site_1, p_site_2, p_site_3, p_site_4, and p_site_5 represent the *C. coli* contamination probabilities for farms, retail markets, wholesale markets, slaughterhouses, and slaughter points, respectively. The parameter intercept_bird represents the baseline probability of a broiler testing positive for *C. coli* at a contaminated site, assuming no effect from site type and province of supply areas. The fixed-effect coefficients beta_ret, beta_slh, beta_slp, and beta_who quantify the effects of retail markets, slaughterhouses, slaughter points, and wholesale markets on the probability of a broiler testing positive for *C. coli* at a contaminated site, respectively. The fixed-effect coefficients beta_cbac, beta_chdqn, beta_chon, and beta_cot quantify the effects of supply areas—Bac Giang, Hai Duong/Quang Ninh, Ha Noi, and other provinces—on the probability of a broiler testing positive for *C. coli* at a contaminated site, respectively. The parameter sigma_1 means the variance of the random intercept.
(TIFF)

**S4 Fig. Pairs plot with four chains for the selected ideal *C. jejuni* model.** This figure shows the pairs plot of the selected *C. jejuni* model, which showed the distribution of each parameter's posterior values and its interactions with other parameters. The parameters p_site_1, p_site_2, p_site_3, p_site_4, and p_site_5 represent the *C. jejuni* contamination probabilities for farms, retail markets, wholesale markets, slaughterhouses, and slaughter points, respectively. The parameter intercept_bird represents the baseline probability of a broiler testing positive for *C. jejuni* at a contaminated site, assuming no effect from site type. The fixed-effect coefficients beta_ret, beta_who, beta_slh, and beta_slp quantify the effects of retail markets, wholesale markets, slaughterhouses, and slaughter points on the probability of a broiler testing positive for *C. jejuni* at a contaminated site, respectively. The parameter sigma_1 means the variance of the random intercept.
(TIFF)

**S5 Fig. Post-predictive check for the selected *C. coli* model: The observed (solid circle shape) and predicted (triangle shape) proportion of broiler samples testing positive by different sites and from different catchment provinces for *C. coli*.**
(TIFF)

**S6 Fig. Post-predictive check for the selected *C. jejuni* model: The observed (solid circle shape) and predicted (triangle shape) proportion of broiler samples testing positive by different sites for *C. jejuni*.**
(TIFF)

**S7 Fig. Trace plots with four chains for the selected ideal NTS model.** This figure shows the trace plots (n chains = 4, n iteration = 50000) of the selected NTS model, which displayed tight, consistent horizontal bands, indicating no visual signs of non-convergence within the chains. The fixed-effect coefficient beta_end quantifies the effect of endpoints on the probability of a site being contaminated with NTS. The parameter intercept_site represents the baseline NTS contamination probability of a site when site type has no effect.
(TIFF)

**S8 Fig. Pairs plot with four chains for the selected ideal NTS model.** This figure shows the pairs plot of the selected NTS model, which showed the distribution of each parameter's posterior values and its interactions with other parameters.

The fixed-effect coefficient beta_end quantifies the effect of endpoints on the probability of a site being contaminated with NTS. The parameter intercept_site represents the baseline NTS contamination probability of a site when site type has no effect.
(TIFF)

**S9 Fig. The observed (solid circle shape) and predicted (triangle shape) proportions of environmental samples testing positive by farms, endpoints and all sites for NTS.**
(TIFF)

**S1 Data. Three datasets with their variable dictionaries containing the data used in this manuscript.** A document 'README.text' describes the structure of the datasets.
(ZIP)

## Acknowledgments

We would like to thank all the people who helped with field collection, sample archiving, and analysis. We thank the research team from the National Institute of Veterinary Research (NIVR), Vietnam, who assisted with the fieldwork, in particular with sample collection and laboratory analysis. We thank the farmers, sellers, and all the many other actors of the productions distribution networks who took part in the study.

## Author contributions

**Conceptualization:** Thuy Thi Hoang, Hoa Thi Thanh Pham, Mathew Hennessey, Ngoc Thi Pham, Guillaume Fournié, Anne Conan.

**Data curation:** Chen Xin, Thuy Thi Hoang, Chun Ting Angus Lam, Guillaume Fournié, Anne Conan.

**Formal analysis:** Chen Xin, Saira Butt, Mathew Hennessey, Guillaume Fournié, Anne Conan.

**Funding acquisition:** Patricia Lynne Conway, Richard A. Stabler, Damer Blake, Fiona Tomley, Dirk Pfeiffer, Ngoc Thi Pham, Guillaume Fournié.

**Investigation:** Thuy Thi Hoang, Duong Quy Truong, Nhat Thi Tran, Son Thi Thanh Dang, Huong Quynh Luu, Nhung Thi Hong Le, Hang Thi Thu Tran, Kelyn Seow, Burhan Lehri.

**Methodology:** Chen Xin, Hoa Thi Thanh Pham, Saira Butt, Kelyn Seow, Patricia Lynne Conway, Richard A. Stabler, Damer Blake, Ngoc Thi Pham, Guillaume Fournié, Anne Conan.

**Project administration:** Hoa Thi Thanh Pham, Damer Blake, Fiona Tomley, Dirk Pfeiffer, Ngoc Thi Pham, Guillaume Fournié.

**Resources:** Duong Quy Truong, Huong Quynh Luu, Patricia Lynne Conway, Richard A. Stabler, Dirk Pfeiffer, Ngoc Thi Pham.

**Software:** Chen Xin, Saira Butt, Mathew Hennessey, Guillaume Fournié, Anne Conan.

**Supervision:** Priscilla F. Gerber, Guillaume Fournié, Anne Conan.

**Validation:** Guillaume Fournié.

**Visualization:** Chen Xin, Guillaume Fournié, Anne Conan.

**Writing – original draft:** Chen Xin, Guillaume Fournié, Anne Conan.

**Writing – review & editing:** Chen Xin, Thuy Thi Hoang, Duong Quy Truong, Nhat Thi Tran, Hoa Thi Thanh Pham, Son Thi Thanh Dang, Huong Quynh Luu, Nhung Thi Hong Le, Saira Butt, Hang Thi Thu Tran, Kelyn Seow, Mathew Hennessey, Burhan Lehri, Chun Ting Angus Lam, Priscilla F. Gerber, Patricia Lynne Conway, Richard A. Stabler, Damer Blake, Fiona Tomley, Dirk Pfeiffer, Ngoc Thi Pham, Guillaume Fournié, Anne Conan.

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
