## [Decision Letter · Decision Letter 0]

5 Jun 2025

Prevalence of Campylobacter and non-typhoidal Salmonella along broiler chicken production and distribution networks, Vietnam

Dear Dr. Conan,

Thank you for submitting your manuscript to PLOS Neglected Tropical Diseases. After careful consideration, we feel that it has merit but does not fully meet PLOS Neglected Tropical Diseases's publication criteria as it currently stands. Therefore, we invite you to submit a revised version of the manuscript that addresses the points raised during the review process.

Please submit your revised manuscript within 60 days Aug 04 2025 11:59PM. If you will need more time than this to complete your revisions, please reply to this message or contact the journal office at plosntds@plos.org. Please include the following items when submitting your revised manuscript:

We look forward to receiving your revised manuscript.

Kind regards,

Francesca Schiaffino, DVM PhD

Guest Editor

Ana LTO Nascimento

Section Editor

Shaden Kamhawi

co-Editor-in-Chief

Paul Brindley

co-Editor-in-Chief

**Additional Editor Comments:**

Dear Dr. Cohen,

Thank you for submitting your manuscript "Prevalence of Campylobacter and non-typhoidal Salmonella along broiler chicken production and distribution networks, Vietnam". The reviewers considered the work valuable, yet requiring major revisions before it is considered for publication. Please see the comments below and modify the manuscript accordingly.

In addition to what the reviewers have noted, I would like to highlight the following major themes:

- Organization of the methods section of the manuscript. Please make sure that the selection of the province wards, sampling points and poultry samples are explained and justified. Currently, its overall cumbersome to understand the selection and inclusion process.

- The methods mention the use of an informed consent. Please incorporate an Ethics section in the manuscript.

- Please explain how many colonies compatible with Campylobacter and NTS were selected for processing. Please explain how the quality of the NTS genomes was assessed and where are genomes made available to the public.

- The methods related to the bioinformatic analysis of NTS is limited. Major editions are required to ascertain the analysis process is reproducible (with reads and genomes made publicly available). For example, what CGE tools were utilized and for what purpose?

- qPCR methods are missing. Please specify Campylobacter genus primer stated in methods but not visualized in table.

- Results, Discussion, Conclusion and Table/Figure concerns have already been highlighted by the reviewers.

We look forward to reviewing a new version of your work. 

Thank you.

**Journal Requirements:**

1) <carina-action-element class="ng-star-inserted">Please ensure that the CRediT author contributions listed for every co-author are completed accurately and in full.</carina-action-element> 

<carina-action-element class="ng-star-inserted">At this stage, the following Authors/Authors require contributions: </carina-action-element><carina-action-element class="ng-star-inserted">Nhung Thi Hong Le</carina-action-element><carina-action-element class="ng-star-inserted">. Please ensure that the full contributions of each author are acknowledged in the "Add/Edit/Remove Authors" section of our submission form.</carina-action-element> <carina-action-element class="ng-star-inserted">The list of CRediT author contributions may be found here: https://journals.plos.org/</carina-action-element><carina-action-element class="ng-star-inserted">plosntds</carina-action-element><carina-action-element class="ng-star-inserted">/s/authorship#loc-author-contributions</carina-action-element>  2) <carina-action-element class="ng-star-inserted">We do not publish any copyright or trademark symbols that usually accompany proprietary names, eg ©, ®, or TM (e.g. next to drug or reagent names). Therefore please remove all instances of trademark/copyright symbols throughout the text, including:</carina-action-element> <carina-action-element class="ng-star-inserted">- ® on page: 8.</carina-action-element>  3) <carina-action-element class="ng-star-inserted">Some material included in your submission may be copyrighted. According to PLOSu2019s copyright policy, authors who use figures or other material (e.g., graphics, clipart, maps) from another author or copyright holder must demonstrate or obtain permission to publish this material under the Creative Commons Attribution 4.0 International (CC BY 4.0) License used by PLOS journals. Please closely review the details of PLOSu2019s copyright requirements here: PLOS Licenses and Copyright. If you need to request permissions from a copyright holder, you may use PLOS's Copyright Content Permission form.</carina-action-element> <carina-action-element class="ng-star-inserted">Please respond directly to this email and provide any known details concerning your material's license terms and permissions required for reuse, even if you have not yet obtained copyright permissions or are unsure of your material's copyright compatibility. Once you have responded and addressed all other outstanding technical requirements, you may resubmit your manuscript within Editorial Manager. </carina-action-element> <carina-action-element class="ng-star-inserted">Potential Copyright Issues:</carina-action-element> <carina-action-element class="ng-star-inserted">i) Figure Figure 1. Please (a) provide a direct link to the base layer of the map (i.e., the country or region border shape) and ensure this is also included in the figure legend; and (b) provide a link to the terms of use / license information for the base layer image or shapefile. We cannot publish proprietary or copyrighted maps (e.g. Google Maps, Mapquest) and the terms of use for your map base layer must be compatible with our CC BY 4.0 license. Note: if you created the map in a software program like R or ArcGIS, please locate and indicate the source of the basemap shapefile onto which data has been plotted. If your map was obtained from a copyrighted source please amend the figure so that the base map used is from an openly available source. Alternatively, please provide explicit written permission from the copyright holder granting you the right to publish the material under our CC BY 4.0 license. If you are unsure whether you can use a map or not, please do reach out and we will be able to help you. The following websites are good examples of where you can source open access or public domain maps: * U.S. Geological Survey (USGS) - All maps are in the public domain. (http://www.usgs.gov) * PlaniGlobe - All maps are published under a Creative Commons license so please cite u201cPlaniGlobe, http://www.planiglobe.com, CC BY 2.0u201d in the image credit after the caption. (http://www.planiglobe.com/?lang=enl) * Natural Earth - All maps are public domain. (http://www.naturalearthdata.com/about/terms-of-use/).</carina-action-element> 4) <carina-action-element class="ng-star-inserted">Please </carina-action-element><carina-action-element class="ng-star-inserted">amend your</carina-action-element><carina-action-element class="ng-star-inserted"> detailed Financial Disclosure statement. This is published with the article. It must therefore be completed in full sentences and contain the exact wording you wish to be published.</carina-action-element> <carina-action-element class="ng-star-inserted">State the initials, alongside each funding source, of each author to receive each grant. For example: "This work was supported by the National Institutes of Health (####### to AM; ###### to CJ) and the National Science Foundation (###### to AM)." 2) State what role the funders took in the study. If the funders had no role in your study, please state: "The funders had no role in study design, data collection and analysis, decision to publish, or preparation of the manuscript.".</carina-action-element> <carina-action-element class="ng-star-inserted">If you did not receive any funding for this study, please simply state: u201cThe authors received no specific funding for this work.u201d</carina-action-element>  

**Reviewers' Comments:**

Reviewer's Responses to Questions

**Key Review Criteria Required for Acceptance?**

**Methods:**

-Are the objectives of the study clearly articulated with a clear testable hypothesis stated?

-Is the study design appropriate to address the stated objectives?

-Is the population clearly described and appropriate for the hypothesis being tested?

-Is the sample size sufficient to ensure adequate power to address the hypothesis being tested?

-Were correct statistical analysis used to support conclusions?

-Are there concerns about ethical or regulatory requirements being met?

Reviewer #1: Are the objectives of the study clearly articulated with a clear testable hypothesis stated?

Partially. While the study’s objectives are generally described, they are not fully comprehensive. Specifically, the molecular characterization of isolates via sequencing—an important component of the study—is not included in the stated objectives at the end of the Introduction. The authors should revise the objectives to fully capture the scope of the work and clearly articulate the testable hypothesis.

Is the study design appropriate to address the stated objectives?

Yes. The study employs a cross-sectional design with sampling across multiple stages of the broiler chicken supply chain. This is appropriate for determining prevalence and identifying risk factors along the production and distribution network. However, some aspects of the site selection and stratification require clarification to fully understand the sampling logic and ensure representativeness.

Is the population clearly described and appropriate for the hypothesis being tested?

Partially. The population under study—broiler chickens from different supply chain stages in Vietnam—is appropriate. However, details about the number and distribution of sites per type (e.g., slaughterhouses, markets) are not clearly presented in the Methods section and only become evident later in the results and tables. This could lead to confusion about how the sampling population was defined and selected.

Is the sample size sufficient to ensure adequate power to address the hypothesis being tested?

Unclear. While the study includes a large number of samples and reports meaningful prevalence estimates, no justification for the sample size or power calculation is provided. Including this information would strengthen the methodological rigor and support the validity of the findings.

Were correct statistical analyses used to support conclusions?

Yes, with minor concerns. The use of Bayesian models and molecular analyses is appropriate and robust. However, the order of presentation of the methods does not align with the results, which can make interpretation difficult. Reorganizing the methods to match the results flow would improve clarity.

Are there concerns about ethical or regulatory requirements being met?

No major concerns noted. However, the manuscript would benefit from more details about sample transport and processing conditions (e.g., average time from collection to processing), particularly to ensure confidence in microbiological integrity. Explicit mention of ethics approvals or permissions for sampling animals and accessing facilities should also be verified for completeness.

Reviewer #2: This manuscript addresses a relevant and underexplored public health issue, the distribution of foodborne pathogens, Campylobacter spp. and non-typhoidal Salmonella (NTS), along poultry production and distribution networks (PDNs) in Vietnam. The study employs a well-designed cross-sectional framework, includes key production nodes, and utilizes whole genome sequencing and Bayesian hierarchical models for prevalence estimation. These strengths position the manuscript as a valuable contribution to the field of One Health and food safety surveillance in low- and middle-income countries (LMICs). However, several methodological and analytical steps require further refinement and elaboration before publication:

1. Sample or sample size justification is missing a rationale or calculation for the sample size (102 sites, 565 broilers) is not included. While the stratification across farms and endpoints is appreciated, it is unclear whether the study was powered to detect meaningful differences between site types, provinces, or broiler types. A formal justification — whether based on expected prevalence, confidence intervals, or DIC stability in Bayesian modeling — would improve methodological rigor.

2. Differences of sampling between pathogens: Campylobacter was detected from cecal content of birds, while NTS was identified from environmental samples. The could mislead interpretations when comparing prevalences, this limitations should be assessed explicitly.

3. It is not clear why WGS was not performed for Campylobacter: the lack of this data is an important limitation. Authors should clarify whether Campy isolates were excluded from sequencing due to logistical reasons, and justify in methods.

4. Limited data for farm contexts for modeling: Bayesian models incorporate site type and provincial location but omit farm-level variables that could be critical in explaining heterogeneity (density, antibiotic use, etc.). If such data were unavailable, this should be clearly stated (and addressed in limitations). If collected but not used, authors should explain the reason for exclusion in the model.

**Results**

-Does the analysis presented match the analysis plan?

-Are the results clearly and completely presented?

-Are the figures (Tables, Images) of sufficient quality for clarity?

Reviewer #1: Does the analysis presented match the analysis plan?

Partially. The analyses performed are consistent with what is described in the Methods section (e.g., prevalence estimation, molecular characterization, Bayesian analysis). However, the order in which the analyses are presented in the Results does not match the order in the Methods, which affects clarity. For example, molecular analysis appears first in the Methods, but is reported after the Bayesian results in the Results section. Aligning the order between sections would improve readability.

Are the results clearly and completely presented?

Partially. While the overall findings are presented with adequate detail, some elements are unclear or not fully explained. For instance, the total number of slaughterhouses and other site types sampled is not clearly described until readers reach Table 2. This should be clarified earlier in the text or in a summary table. Additionally, the organization of some results could be improved for transparency.

Are the figures (Tables, Images) of sufficient quality for clarity?

No, revisions are needed. Several tables require reformatting for clarity:

Table 3 is confusing, as the categories are not mutually exclusive and the mix of fast- and slow-growing chickens introduces an additional layer that should be subdivided for clarity. The use of asterisks is also redundant with the caption and would benefit from clearer explanation.

Table 1 includes redundant information in the "Target" and "Product size" columns and should be simplified for better understanding.

Overall, tables need formatting adjustments and clearer labeling to enhance their interpretability.

Reviewer #2: Authors present detailed descriptive and model-based findings on the prevalence of Campylobacter spp. and non-typhoidal Salmonella (NTS) across different poultry production and distribution sites in northern Vietnam. While the structure is clear and the use of Bayesian high density intervals is appropriate, there are several areas that require clarification:

1. The manuscript frequently alternates between reporting crude (observed) prevalence and modeled posterior estimates without always making the distinction clear. This could lead to confusion (for posterior less familiar readers with Bayesian models). It would help to clearly label whether a prevalence figure is from raw data or model output in both text and tables.

2. Integration of genomic data: I strongly suggest a deeper genomic analysis (only NTS serotype diversity was summarized). Authors should align with their aims why more extensive genomic analysis was not performed. This should be clarified in the discussion and considered a missed opportunity.

3. Model interpretations are not detailed: authors state that convergence was achieved and effective sample sizes were sufficient, but diagnostic results (e.g., trace plots, Rhat, PPC summaries) are only described in supplementary materials. Including a brief narrative interpretation will help.

**Conclusions**

-Are the conclusions supported by the data presented?

-Are the limitations of analysis clearly described?

-Do the authors discuss how these data can be helpful to advance our understanding of the topic under study?

-Is public health relevance addressed?

Reviewer #1: Are the conclusions supported by the data presented?

Yes. The conclusions are generally supported by the data presented in the Results. The study finds important prevalence data on Campylobacter and non-typhoidal Salmonella in the broiler chicken production and distribution networks in Vietnam. However, some clarifications on the analysis process (particularly regarding site selection and sampling strategy) would further strengthen the connection between the data and the conclusions.

Are the limitations of the analysis clearly described?

Partially. While the study provides a thorough analysis, the limitations are not fully addressed. The manuscript would benefit from a more explicit discussion of potential sampling biases (e.g., selection bias in site types or sample transport conditions), methodological limitations (e.g., the possibility of false negatives/positives in pathogen detection), and generalizability of the findings. A clear mention of limitations would provide a more balanced perspective on the data and its implications.

Do the authors discuss how these data can be helpful to advance our understanding of the topic under study?

Reviewer #2: The conclusions occasionally overstate the strength of the evidence, especially where the data were limited. For example, suggesting widespread policy changes based on small sample sizes in slaughterhouses (n=6) and wholesale markets (n=10) should be more cautiously framed.

There is no mention of limitations in the genomic analysis, particularly the lack of WGS for Campylobacter, which should temper the generalizability of genomic insights.

The implications for human health are mentioned, but not linked to actual data (e.g., no integration with human case burden or clinical surveillance in Vietnam). This disconnect should be acknowledged.

In summary:

1. Conclusions are partially supported by the data: some conclusions extend beyond the strength of the evidence presented, the recommendation for broad-scale policy or surveillance interventions would be more convincing if supported by more robust genomic data (particularly for Campylobacter) or demonstrated links to human health outcomes. Generalizability to all of Vietnam should be adjusted, especially given the limited number of slaughterhouses and wholesale markets sampled.

2. Limitations of analysis are not fully described. While some are acknowledged, several important ones are underreported or omitted, including: asymmetric sampling approach between pathogens, absence for WGS data for Campy

3. The manuscript acknowledges the importance of evaluating pathogen prevalence at multiple nodes in PDNs, which contributes the understanding of where contamination risks are concentrated. The authors also note the utility of molecular methods for surveillance in resource-limited settings. However, genomic component is underdeveloped, the study could have advanced understanding of Salmonella epidemiology more deeply through comparative genomics, resistance gene analysis, or clustering. For Campylobacter, the potential to advance understanding is not fully realized given the absence of WGS or deeper typing beyond species identification.

4. The manuscript appropriately frames Campylobacter and NTS as foodborne pathogens of global health concern, and highlights the potential for transmission along poultry value chains. The conclusion references the need for improved hygiene and regulation at slaughter and retail points.

**Editorial and Data Presentation Modifications?**

Reviewer #1: (No Response)

Reviewer #2: Minor revision.

**Summary and General Comments**

Reviewer #1: The manuscript titled "Prevalence of Campylobacter and non-typhoidal Salmonella along broiler chicken production and distribution networks, Vietnam" presents a well-executed and scientifically relevant study that contributes valuable insights to the global understanding of foodborne pathogens in poultry production chains. The findings are important for both the scientific community and public health authorities, particularly in the context of food safety in low- and middle-income countries. The methodology is generally sound, and the data are robust. However, several minor issues should be addressed to improve the clarity and transparency of the manuscript.

Minor Comments for Revision

Objectives in the Introduction

The stated objectives at the end of the Introduction appear incomplete. While the study includes molecular characterization through sequencing, this important component is not mentioned in the stated aims. It is recommended that the authors revise the objectives to fully reflect all key aspects of the study.

Site Selection Description

The subsection "Site selection" in the Materials and Methods is somewhat confusing and lacks alignment with the first section of the Results. For instance, lines 149–150 mention:

"All wholesale markets (N = 10) and slaughterhouses (N = 6) that processed slow-growing broiler chickens in the four study provinces were recruited,"

yet the total number of slaughterhouses sampled remains unclear until Table 2, where 18 positive samples are reported. A clearer explanation of how many slaughterhouses and other site types were evaluated should be included upfront. Consider providing a descriptive table in the Methods section to summarize the distribution and number of sites per type, and clarify whether any stratification approach was applied in the site selection process.

Method-Results Alignment

The order of methodological descriptions does not match the order of presentation in the Results section. For example, molecular analyses are presented before Bayesian analysis in the Methods, but this order is reversed in the Results. It is recommended to align the structure of the Methods with the flow of the Results to improve readability.

Table 3 Clarity and Formatting

Table 3 is difficult to interpret. The categories presented are not mutually exclusive, and the combination of fast- and slow-growing chicken types introduces an additional layer of categorization that could benefit from clearer subdivision. It appears that the table format may not fully conform to journal standards, and a reorganization could greatly improve comprehension. The asterisk notation used in the table is redundant with the caption and would be more useful if better integrated into a clearer explanation within the caption.

Table 1 Simplification

Table 1 also requires formatting improvements. The columns “Target” and “Product size” contain redundant information that could be simplified for clarity.

Sample Processing Time (Line 177)

The manuscript states that "Cecal content samples were processed upon arrival at the laboratory of the National Institute of Veterinary Research." It is important to clarify the average time between sample collection and processing, and whether there was significant variability among samples, as this can influence the reliability of microbial detection.

Reviewer #2: This manuscript presents a cross-sectional, multi-site study investigating the prevalence of Campylobacter spp. and non-typhoidal Salmonella (NTS) along poultry production and distribution networks (PDNs) in Vietnam.

However, there are several important limitations that reduce the overall impact and scientific value of the work in its current form.

1. Limited genomic analysis: WGS was performed on Salmonella isolates but only used for serotyping . There is no analysis of resistance genes, virulence genes, or genomic clustering/phylogeny. These elements are widely expected in genomic surveillance studies and would enhance the public health relevance of the findings. Moreover, Campylobacter isolates were not sequenced, and this gap is not adequately addressed in the text.

2. No explicit hypotheses tested and limited statistical interpretation: the study is descriptive, and although Bayesian modeling is correctly implemented, the lack of posterior probability estimates of differences between categories weakens interpretability. For example, the manuscript does not quantify how likely it is that prevalence differs between site types or between species (Campy coli vs C. jejuni), despite having the framework to do so. Authors should report posterior probabilities of group differences where relevant, clarifying differences supported by data.

3. Some critical limitations are only partially acknowledged. For example, the asymmetric sampling strategy (Campy from cecal samples; Salmonella from environmental swabs) is not discussed, and the limited genomic scope for Campylobacter is omitted entirely. Conclusions imply policy recommendations that are not fully supported by the dataset (e.g., generalizable surveillance implications from very few slaughterhouses sampled).

PLOS authors have the option to publish the peer review history of their article (what does this mean? ). If published, this will include your full peer review and any attached files.

**Do you want your identity to be public for this peer review?** For information about this choice, including consent withdrawal, please see our Privacy Policy .

Reviewer #1: No

Reviewer #2: No

**Figure resubmission:**

**Reproducibility:**



---

## [Decision Letter · Decision Letter 1]

1 Oct 2025

Dear Dr. Conan,

We are pleased to inform you that your manuscript 'Prevalence of Campylobacter and non-typhoidal Salmonella along broiler chicken production and distribution networks, Northern Vietnam' has been provisionally accepted for publication in PLOS Neglected Tropical Diseases.

Best regards,

Francesca Schiaffino, DVM PhD

Guest Editor

Ana LTO Nascimento

Section Editor

Shaden Kamhawi

co-Editor-in-Chief

Paul Brindley

co-Editor-in-Chief

Reviewer's Responses to Questions

**Key Review Criteria Required for Acceptance?**

**Methods**

-Are the objectives of the study clearly articulated with a clear testable hypothesis stated?

-Is the study design appropriate to address the stated objectives?

-Is the population clearly described and appropriate for the hypothesis being tested?

-Is the sample size sufficient to ensure adequate power to address the hypothesis being tested?

-Were correct statistical analysis used to support conclusions?

-Are there concerns about ethical or regulatory requirements being met?

Reviewer #1: (No Response)

Reviewer #2: The authors have substantially improved the manuscript and addressed the concerns in the first review. The objectives are now clearly stated and aligned with the methods, including explicit mention of NTS serotyping. The cross-sectional design across farms, slaughterhouses, and markets in Northern Vietnam is appropriate and the study population is well described. A retrospective Bayesian power analysis was added, clarifying that while the sample size is sufficient for some comparisons, power is limited for others, which is now acknowledged as a limitation. Statistical analyses are appropriate, and clarified in the revision. The lack of Campylobacter WGS remains a limitation, but it is now explicitly acknowledged.

**Results**

-Does the analysis presented match the analysis plan?

-Are the results clearly and completely presented?

-Are the figures (Tables, Images) of sufficient quality for clarity?

Reviewer #1: (No Response)

Reviewer #2: The results presented are consistent with the stated analysis plan. The descriptive findings for Campylobacter and NTS are clearly structured, followed by Bayesian model outputs that align with the methods. The reorganization of the results section improves the flow and makes interpretation easier.

Tables and figures are of sufficient quality for clarity, and they effectively summarize the prevalence estimates, Bayesian outputs, and serotype distributions. They are legible, appropriately labeled, and support the text without redundancy.

**Conclusions**

-Are the conclusions supported by the data presented?

-Are the limitations of analysis clearly described?

-Do the authors discuss how these data can be helpful to advance our understanding of the topic under study?

-Is public health relevance addressed?

Reviewer #1: (No Response)

Reviewer #2: The conclusions are now well aligned with the data. The authors have moderated their claims, restricted generalization to Northern Vietnam, and focused on the main findings of pathogen prevalence, serotyping, and Bayesian modeling. Limitations are clearly acknowledged, including differences in sampling methods, absence of Campylobacter WGS, limited statistical power in some comparisons, and lack of human data.

Overall, the conclusions are supported, the limitations are transparent, and the study’s contribution and relevance are clearly stated.

**Editorial and Data Presentation Modifications?**

Reviewer #1: (No Response)

Reviewer #2: Define “HDI” (highest density interval) at first mention in the text and in table/figure legends for readers unfamiliar with Bayesian terminology.

Ensure that all tables and figures clearly indicate the sample size (n) for each node of the production and distribution (farms, slaughterhouses, markets) in titles or legends.

Clarify data availability: while epidemiological and serotyping data appear to be included in the supplementary materials, genomic datasets are stated to be released with future publications. For consistency with PLOS data policy, it would be advisable to deposit and link these genomic datasets at the time of publication, even if further analyses are reported separately.

**Summary and General Comments**

Reviewer #1: (No Response)

Reviewer #2: This revised manuscript shows clear improvements in clarity and transparency.

The study has several strengths. The design is robust, covering multiple nodes of the production market chain, and the methods are now described in detail and organized in a way that makes them much easier to follow. Ethical approvals and sample handling are clearly documented. The use of Bayesian models is appropriate and well explained, and the results are presented in a clear, logical sequence. Importantly, the authors expanded their discussion of limitations, being open about the differences in sampling strategies between pathogens, the absence of Campylobacter WGS, limited statistical power for some comparisons, and the lack of human health outcome data. This openness strengthens the credibility of the work.

Some weaknesses remain, mainly in the genomic component. The absence of WGS for Campylobacter does limit the depth of inference, and the NTS genomic analysis is somewhat restricted. That said, the authors explained that this aspect will be addressed in separate publications, and since the primary scope of this paper is epidemiological prevalence, this is reasonable.

Overall, this is a significant contribution. The findings have clear public health implications for food safety and regulation, and the work also illustrates the usefulness of Bayesian methods in contexts where sample sizes are limited. The authors have been responsive to reviewer and editor feedback, and the paper is now more rigorous, and transparent.

PLOS authors have the option to publish the peer review history of their article (what does this mean? ). If published, this will include your full peer review and any attached files.

**Do you want your identity to be public for this peer review?** For information about this choice, including consent withdrawal, please see our Privacy Policy .

Reviewer #1: No

Reviewer #2: **Yes: ** Guillermo Salvatierra

---

## [Editor Report · Acceptance letter]

Dear Dr. Conan,

We are delighted to inform you that your manuscript, "Prevalence of Campylobacter  and non-typhoidal Salmonella along broiler chicken production and distribution networks, Northern Vietnam," has been formally accepted for publication in PLOS Neglected Tropical Diseases.

Best regards,

Shaden Kamhawi

co-Editor-in-Chief

Paul Brindley

co-Editor-in-Chief
